

# Effect of horizontal resolution in North Atlantic mixing and ocean circulation in the EC-Earth3P HighResMIP simulations

Eneko Martin-Martinez[1,2], Amanda Frigola[1], Eduardo Moreno-Chamarro[1,3], Daria Kuznetsova[1],
Saskia Loosveldt-Tomas[1], Margarida Samsó Cabré[1], Pierre-Antoine Bretonnière[1], and Pablo Ortega[1]

[1]Barcelona Supercomputing Center (BSC), Barcelona, Spain
[2]Departament de Dinàmica de la Terra i l'Oceà, Facultat de Ciències de la Terra, Universitat de Barcelona (UB), Barcelona, Spain
[3]now at Max Planck Institute for Meteorology, Hamburg, Germany

**Correspondence:** Eneko Martin-Martinez (eneko.martin@bsc.es)

**Abstract.**

We investigate the impact of increasing horizontal model resolution on the oceanic mixing processes in the North Atlantic, their drivers, their link with the Atlantic Meridional Overturning Circulation (AMOC), and the propagation of newly generated dense waters through the deep western boundary current (DWBC). We use three versions of the EC-Earth Earth System Model, one of standard resolution (SR, $\sim 1°$ in the ocean), one of high resolution (HR, $\sim 0.25°$ in the ocean) and one of very high resolution (VHR, $\sim 1/12°$ in the ocean). The higher resolutions allow for the explicit simulation of mesoscale processes that are parametrized at the coarse resolution, with additional improvements in ocean topography, boundary currents and air-sea interactions.

We find that the North Atlantic Oscillation plays a critical role in driving the mixed layer depth (MLD) in the Labrador Sea at HR and VHR. The three resolutions also show the influence of surface salinity signals in the mixing, with the VHR configuration showing a distinct slow propagation of these signals from the eastern subpolar gyre into the Labrador Sea. Furthermore, March MLD shows a strong positive bias in HR, which is reduced in VHR. In terms of the AMOC, resolution plays a pivotal role in shaping its response to the mixing. At the higher resolutions, the signal of the newly formed dense waters propagates faster along the better-resolved boundary current, indicating a shift from advective propagation to wave propagation of the signals. Additionally, the persistence of the AMOC responses to MLD is much shorter in VHR (less than 2 years) than for SR and HR, which exhibit longer-lived changes. These differences highlight how resolution affects both the timing and spatial reach of the AMOC changes.

Our study underscores the importance of model resolution in accurately simulating the North Atlantic's oceanic processes and their implications for the AMOC. While the VHR configuration offers a more realistic climatology of the Labrador Sea MLD, the results also demonstrate significant differences in variability and persistence across resolutions. These findings stress the need for high-resolution simulations to improve the understanding of deep ocean processes and their connection to larger climate systems, although they also highlight challenges in comparing simulated and observed data, particularly given the sparse historical observations and the lack of decadal variability in the model simulations.



## 1 Introduction

Deep water mixing is a key driving process for the Atlantic meridional overturning circulation (AMOC; Kuhlbrodt et al., 2007). At mid-latitudes, the Gulf Stream transports warm and salty waters into the subpolar North Atlantic, where processes such as brine rejection by sea ice formation (Worster and Rees Jones, 2015; Lake and Lewis, 1970) or heat-loss to the atmosphere aloft (Béranger et al., 2010; Pennelly and Myers, 2021) lead to increased seawater surface density. These changes in surface density can break the vertical stratification, resulting in a deeper mixed layer. These processes are particularly relevant during winter when the atmosphere is at its coldest, and the mixed layer increases, reaching its annual maximum at the end of the season (Schiller and Ridgway, 2013). The mixed layer depth (MLD) is typically used to characterise deep water formation in key regions of the North Atlantic Ocean, such as the Irminger and Labrador seas (Koenigk et al., 2021; Ortega et al., 2021). Deep dense anomalies formed in those regions through mixing are later transported along the deep western boundary current (DWBC), modifying the zonal density gradient as they move southwards, which triggers a response of the AMOC overturning circulation via thermal wind balance (Stammer et al., 1999; Ortega et al., 2017a).

Mixing processes and, thus, the AMOC, are directly and indirectly impacted by several mesoscale processes, like mesoscale eddies. These structures have a length scale of $100\,\mathrm{km}$ or smaller, lasting from weeks to months. Their role in North Atlantic variability is essential as they decisively contribute to the transport of water of different properties like temperature and salinity (Volkov et al., 2008; Treguier et al., 2014; Dong et al., 2014), and by extension, to deep mixing in key regions like the Labrador Sea. Understanding and addressing the limitations of climate models that cannot resolve these processes is critical for improving their accuracy and gaining confidence in future climate projections.

The typical horizontal scale resolved by common climate models does not include mesoscale processes. Most models contributing to the Coupled Model Intercomparison Project phase 6 (CMIP6) (Eyring et al., 2016) have an ocean resolution of approximately 1º on mid-latitudes, which corresponds to about $100\,\mathrm{km}$. To resolve the largest mesoscale eddies, model resolution should be finer than the first Rossby radius of deformation. This corresponds to a resolution of at least 1/4º in the equator, about 1/12º in the mid-latitudes and 1/25º in the polar regions (Hallberg, 2013). Therefore, models with 1º grid spacing cannot resolve mesoscale eddies and need to parametrise their contributions (Haarsma et al., 2016; Hallberg, 2013); for that reason, they are also known as eddy-parametrised models. However, such parametrisations are approximations intended to replace the interactions and feedbacks of mesoscale dynamics. This leads to systematic biases in models, including non-realistic convection in the North Atlantic Ocean (Heuzé, 2021), among others.

The High Resolution Model Intercomparison Project (HighResMIP) defined a protocol to investigate the impact of enhancing the resolution in the ocean and the atmosphere (Haarsma et al., 2016). Within this protocol, high-resolution configurations must have a minimum ocean resolution of 0.25°, which is fine enough to resolve some oceanic mesoscale dynamics in the Tropics. Models run at this resolution cannot represent the eddies at higher latitudes, where the Rossby radius of deformation is smaller (Hallberg, 2013), and are typically known as eddy-present or eddy-permitting models.





Recent supercomputing power and model performance improvements have enabled some groups to advance in global coupled modelling, contributing to HighResMIP with horizontal resolutions representing ocean mesoscale eddies up to about 50°
latitudes. These models, commonly known as eddy-resolving or eddy-rich models, usually have a resolution of 1/10º or 1/12º,
corresponding to about 10 km at mid-latitudes. One such model configuration is EC-Earth3P-VHR (Moreno-Chamarro et al.,
2024), developed at the Barcelona Supercomputing Center in the Horizon 2020 PRIMAVERA project.

Several studies have shown that effectively resolving the mesoscale, added to a better representation of the topography
with the increased resolution, reduces long-standing model biases in the ocean (Marzocchi et al., 2015; Menary et al., 2015;
Roberts et al., 2019; Ding et al., 2022; Athanasiadis et al., 2022), tends to deepen the mixed layer in the Labrador Sea (Koenigk
et al., 2021) and improves air-sea interactions (Moreno-Chamarro et al., 2021; Bellucci et al., 2021; Roberts et al., 2020) when
comparing to eddy-parametrised and eddy-permitting models. Resolving the ocean mesoscale also has an important impact on
the interior–boundary currents exchange in the Labrador Sea (Georgiou et al., 2020). To our knowledge, no study to date has
addressed whether and how the ocean resolution affects the drivers of deep water formation and its ultimate link to the AMOC.

Many studies advocate for a leading role of the Labrador Sea in the formation of dense waters through mixing (Roberts
et al., 2020; Yeager et al., 2021; Swingedouw et al., 2022). Some of the processes that drive Labrador Sea mixing are the direct
atmospheric forcing (i.e. via local air-sea exchanges), and density anomalies arriving from the Irminger Sea (Menary et al.,
2020; Petit et al., 2020; Jackson and Petit, 2023; Petit et al., 2023a), or from Arctic outflow waters (Ortega et al., 2017b).
However, it is unclear which ones are the actual key drivers, as the associated studies mix different model setups (e.g. forced
ocean-only vs coupled) and consider resolutions that might be missing essential feedbacks and fine-scale interactions between
the atmosphere and the ocean. Ultimately the influence of North Atlantic mixing on the large-scale AMOC is pre-conditioned
by the ocean mean state, e.g. via local stratification (Lin et al., 2023; Kim et al., 2023; Patrizio et al., 2023; Jackson et al.,
2020), which is sensitive to ocean resolution (Koenigk et al., 2021; Petit et al., 2023b).

To study whether and how resolving mesoscales processes affects the representation of mixing processes and their link with
the AMOC, this study uses HighResMIP simulations with three different versions of the CMIP6 model EC-Earth-3P, based
on the eddy-parameterised, eddy-permitting, and eddy-rich configurations developed for HighResMIP. Section 2 goes through
the methods and describes the model configuration (Section 2.1), the observational data used as a reference (Section 2.2), the
definition of the overturning stream function (Section 2.3) and the statistical methods considered (Section 2.4). Section 3 describes the main results along the three different model configurations, structured in three parts. First, Section 3.1 explores how
increased resolution in the atmosphere and the ocean affects the climatological mixing in the North Atlantic, including stratification in the Labrador Sea, the main deep mixing location. Then, Section 3.2 focuses on the main drivers of the MLD of the
Labrador Sea and Section 3.3 investigates the link between the MLD and the AMOC through its influence on the propagation
of density anomalies along the boundary. We wrap all the conclusions in Section 4 and discuss other open questions.





**Table 1.** Ocean and atmospheric grid configuration of each model resolution and their approximate resolution in mid-latitudes.

|     | Ocean/Sea Ice |         | Atmosphere |        |
| --- | ------------- | ------- | ---------- | ------ |
| SR  | ORCA1         | 100 km  | T255       | 80 km  |
| HR  | ORCA025       | 25 km   | T511       | 40 km  |
| VHR | ORCA12        | 8 km    | T1279      | 16 km  |

## 2 Methodology

### 2.1 Experimental set-up

We use the global coupled climate model EC-Earth3P (Haarsma et al., 2020; Moreno-Chamarro et al., 2024), a version of the model specifically developed within the PRIMAVERA project, to contribute to the first phase of HighResMIP (Haarsma et al., 2016), as part of the CMIP6 initiative. This model version uses the atmospheric IFS cy36r4 model, the ocean NEMO model in its version 3.6, and the sea ice model LIM3. To investigate the role of fine-scale processes in the deep-water formation in the North Atlantic, we compare simulations of eddy-parametrised (SR), eddy-permitting (HR), and eddy-rich (VHR)

configurations of the model with approximate grid spacings in the ocean of about 100, 25 and 8 km, respectively (Table 1 shows more information about each configuration; detailed information of the models can be found in Haarsma et al. (2020); Moreno-Chamarro et al. (2024)). Their atmospheric components also feature gradual enhancements in resolution, from about 80 km to 40 and 16 km, respectively. Since our interest is in comparing internal variability in the North Atlantic, we focus on the HighResMIP 1950-control simulations (i.e. with perpetual radiative forcing conditions from 1950) with one member per

resolution.

The HighResMIP protocol sets a minimum duration of 100 years for the 1950-control experiment (Haarsma et al., 2016). Although the lower-resolution experiments extend much longer, due to the high computational costs of the VHR version, the corresponding 1950-control could only be run for a total of 106 years. Therefore, this resolution limits the number of years we consider for all the resolutions, as we prefer to keep comparable setups. Moreover, we discard the first 30 years of all

simulations to avoid the effects of a drift that lingers after the relatively short spin-up period. The 50 years the HighResMIP protocol recommends are insufficient for the model to reach a trend-free state on the ocean surface (Moreno-Chamarro et al., 2024). We therefore analyse 76 years of each simulation. Also, for all the experiments, we use a second-order polynomial detrending to remove residual model drifts that all the models preserve in the deep ocean layers (not shown).

### 2.2 Observational data

We use oceanic temperature and salinity observations to compare the vertical density profiles and MLD variability. In particular, we use the EN.4.2.2 g10 (EN4 Good et al., 2013) version based on Gouretski and Cheng (2020) Mechanical BathyThermo-graph, and Gouretski and Reseghetti (2010) XpendableBathyThermograph corrections. To produce variables that are compa-



rable for both experiments and observations, we compute in both cases the sigma0 and sigma2 potential density anomalies

from monthly means of potential temperature and practical salinity using the TEOS-10 equation (Roquet et al., 2015). Then,

we use the monthly sigma0 outputs thus derived to compute the MLD following the density threshold of $0.03 \, \mathrm{kg \, m^{-3}}$, setting

the reference depth at 10 m (de Boyer Montégut et al., 2004). In addition, to maximise comparability with the simulations, we

select different time ranges in the observations, depending on the target analysis. We take the 21 years centred around 1950

(1940-1960) to compute all climatological values to stay close to the 1950 radiative forcing conditions. When assessing vari-

ability (e.g. with standard deviations and correlations), we instead consider the last 76 years available (1948-2023), so that both

observations and simulations cover comparable timescales. Since observations include forced signals, which are not present in

the simulations, a second-order polynomial is previously removed from the data to focus on the inter-annual variations, which

are mostly internally driven. Sea ice concentration has been taken from March 1940-1960 average from HadISST2 (Rayner

et al., 2003).

### 2.3 Meridional overturning streamfunction

To characterize the AMOC, we compute the meridional overturning streamfunction for the Atlantic basin in the native grid's

y-axis. Note that ORCA grids are almost regular in the 90º S - 45º N range, which means that the computed transport at those

latitudes will be virtually meridional. We define the transport as the cumulative sum, from bottom to top, of the return flow, see

Eq. (1).

$$\psi_z(t, y, z) = - \int_{z'=-H}^{z} \int_{x'=x_W}^{x_E} v(t, x', y, z') \, dx' \, dz' \tag{1}$$

where $x_W$ and $x_E$ are the west and east boundaries of the basin and $H$ is the basin depth.

In the subpolar region, the density profile changes sharply in the longitudinal direction due to the multiple processes occur-

ring in the area (e.g. Arctic outflows, deep convection, western boundary current, subpolar gyre circulation). For this reason,

we also analyse the overturning streamfunction in sigma-space (see Eq. (2)), which is more suitable to capture the contributions

from deep water formation in the subpolar North Atlantic region (Zhang, 2010; Foukal and Chafik, 2024).

$$\psi_\sigma(t, y, \sigma) = - \int_{x'=x_W}^{x_E} \int_{z'=z_{\sigma_{\max}}}^{z_\sigma} v(t, x', y, z') \, dz' \, dx' \tag{2}$$

where $z_{\sigma_{\max}}$ is the depth where $\sigma$ is maximum, in a stable ocean this will be $-H$ as in Eq. (1). $z_\sigma$ is the depth where the density

is equal to $\sigma$. Note that while in Eq. (1) integrals can be rearranged, in Eq. (2), the integral in $z'$ should be done first as there is

a change in the system of reference.

To study the AMOC signal driven by thermohaline changes in vertical ocean mixing, we remove the Ekman transport from

the overturning streamfunction. We compute the total Ekman transport as the west-east integral of the surface wind stress

divided by the Coriolis parameter and the reference density, see Eq, (3). Note that, as the net transport should be equal to 0,

the resulting transport $V$ in the Ekman layer (approximately the first $30\mathrm{m}$) should be compensated by an equal and opposite

transport, which is assumed to happen uniformly throughout the whole water column.





$$V(t, x', y) = -\frac{1}{\rho_0} \int\limits_{x'=x_W}^{x_E} \frac{1}{f(x', y)} \tau_x(t, x', y) \tag{3}$$

where $\rho_0 = 1025\,\mathrm{kg\,m^{-3}}$ is the reference density, $f$ is the Coriolis parameter and $\tau_x$ is the zonal wind stress. The transport should be divided by the cross-section x-z area to recover the average speed, which is integrated as in Eqs. (1)-(2) to get the streamfunction in both z and sigma spaces.

## 2.4 Result evaluation

All the scientific analyses are performed using well-established open-source scientific programming languages and tools. Most
of the analyses are performed directly with Python and ESMValTool v2.11.0 (Righi et al., 2020; Andela et al., 2024a, b), a software package specifically created to facilitate a rigorous evaluation of CMIP simulation outputs that is especially useful to compare multiple models with observational datasets.

We use Pearson's correlation coefficient, $r$, to assess the interrelation between variables in several analyses. The effective number of degrees of freedom, $N_{\mathrm{eff}}$, of a correlation of $N$-length series is computed using the approach proposed by Afyouni
et al. (2019) to take into account the temporal autocorrelation of the data, see Eq. (4).

$$N_{\mathrm{eff}} = N \left( 1 + 2 \sum_{k=1}^{N-1} \frac{N-k}{N} r_{XX,k}\, r_{YY,k} \right)^{-1} \tag{4}$$

where $r_{XX,k}$ and $r_{YY,k}$ are the autocorrelations of each series at lag $k$. Their statistical significance is computed according to a Student's t distribution setting a p-value of 0.05 (i.e. 95 % confidence level) to determine if the correlation is statistically different from 0.

## 3 Results

### 3.1 North Atlantic mixing

We first look at the climatology of the MLD in March, the month it reaches its yearly maximum. In the three model versions (Fig. 1a-c), the Labrador and Irminger Seas appear to have a stronger mixing than the Nordic Seas. The models show qualitative and quantitative differences with each other in the areas with active mixing around the Labrador Sea. The MLD in the Subpolar
Gyre is much more intense in HR (Fig. 1b), where it exceeds 2000 m in a broad region covering both the Irminger and Labrador Seas. By contrast, VHR (Fig. 1c) has a thinner mixed layer climatology, which can locally reach up to 1500 m deep in the Labrador Sea interior. In addition, this resolution has a comparatively shallower mixing South of Cape Farewell and the Irminger Sea. SR (Fig. 1a) shows an even smaller MLD of about 1000 m, concentrated in a small region south of Cape Farewell. This model resolution has, in fact, a positive bias in the sea ice concentration covering the western Labrador Sea (see
the contour lines in Fig. 1), which damps the local air-sea interactions and, subsequently, the mixing. For similar reasons, due




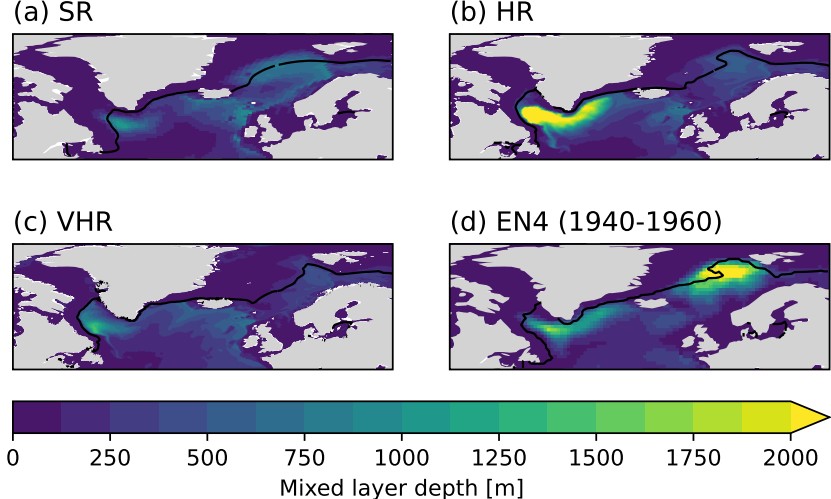

**Figure 1.** March MLD climatology for SR (a), HR (b), VHR (c), and EN4 1940-1960 (d). The black contour line shows the climatological March sea-ice concentration at 15 %, for the same dataset in (a-c), and HadISST2 1940-1960 in (d).

to the positive bias in sea ice concentrations in the Nordic Seas, all the model resolutions simulate very little mixing in that region.

By contrast, EN4 (Fig. 1d) shows very intense mixing in the Nordic Seas region, with much stronger values than in the models, which can locally exceed 2000 m depth. Some of these differences, however, might derive from the large uncertain-
ties in EN4 during the considered period (1940-1960), in which observations in the subsurface were very scarce (Killick, 2021). It could also happen that the way of generating this observation-based product affected the estimated MLD values. EN4 uses spatial and temporal interpolations of different observational sources, including instantaneous profiles, to produce a monthly regularly gridded dataset. This process may artificially reduce the vertical stratification, resulting in a higher MLD (de Boyer Montégut et al., 2004). Nonetheless, EN4 still gives useful qualitative information about the spatial extent and the key
regions of deep water mixing, allowing us to conclude that all model resolutions underestimate the mixing in the Norwegian Seas, with values that are consistently shallower than in the Labrador Sea, unlike in observations.

When looking at the variability of the MLD, as represented by the standard deviation in time, this is generally greater in the regions where the climatological mean values are also large (Fig. 2). In the case of SR and HR, Fig. 2a-b, the standard deviation in the Western Subpolar Gyre is of similar magnitude, reaching maximum values of around 1000 m, although differences are
found in the Nordic Seas, where SR shows higher variability. Compared to SR and HR, VHR shows lower variability in the Labrador Sea, with values of around 400 m and even lower in the Nordic Seas (Fig. 2c). In EN4, for the 1948-2023 period (Fig. 2d), we see consistently stronger variability than in the models (i.e. values above 1300 m) in all regions. The substantially higher observational uncertainty in the early part of EN4 compared to the present period may affect the associated MLD variability. We also note that EN4-derived data might include forced signals that have not been properly removed by the
polynomial detrending. These include low-frequency modulations not present in the (unforced) control simulations, as well as



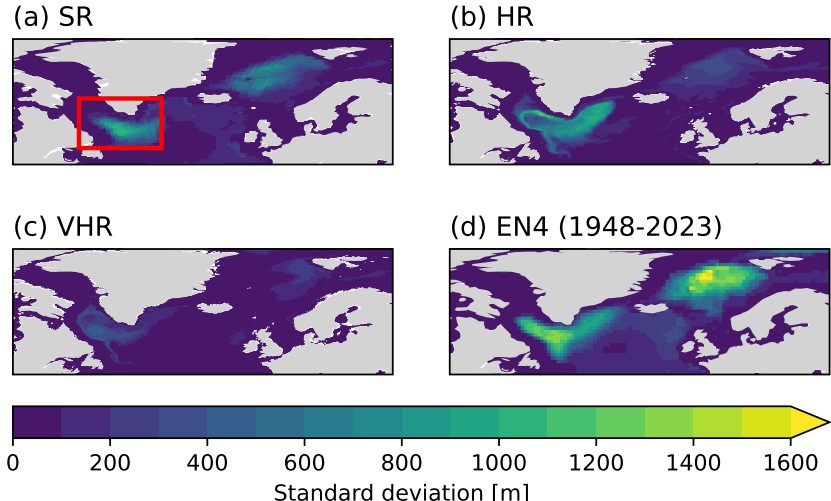

**Figure 2.** March standard deviation of MLD for SR (a), HR (b), VHR (c), and EN4 1948-2023 (d). The box used to compute MLD time-series and vertical profiles is shown in (a).

**Table 2.** Correlation (corr) and root-mean-square error (rmse) of the climatological vertical profiles from Fig. 3a-c of three model resolutions with respect to EN4. All the correlations are significant with a p-value $\ll 0.05$. Results with the best performance, i.e. higher correlation and lower root-mean-square error, are highlighted in bold.

| | Density | | Salinity | | Temperature | |
|---|---|---|---|---|---|---|
| | corr | rmse $(\mathrm{kg\,m^{-3}})$ | corr | rmse (psu) | corr | rmse (°C) |
| SR | 0.976 | **0.035** | 0.881 | 0.111 | 0.852 | 0.907 |
| HR | 0.967 | 0.050 | 0.866 | 0.056 | 0.960 | 0.425 |
| VHR | **0.987** | 0.048 | **0.969** | **0.049** | **0.986** | **0.217** |

some additional interannual variability linked to the response to the three major volcanic eruptions that occurred in 1963, 1982 and 1991.

We now focus on the Labrador Sea, where the models show important MLD differences, and where MLD has extensively been linked to changes in the AMOC (Ortega et al., 2021; Yeager et al., 2021; Koenigk et al., 2021; Lin et al., 2023). To understand the differences in the MLD mean state and variability across the three model resolutions and observations, we examine their vertical density stratification, which is a key preconditioner for mixing. Figure 3a shows the climatological vertical profiles of density in March for the Labrador Sea, computed as area-weighted averages in the box 60º W-35º W 50º N-65º N (from Ortega et al., 2021), also shown with a red box in Fig. 2a. Labrador Sea density is found to be more stratified in SR than in EN4, and less stratified in HR and VHR. We use two different complementary metrics to measure the degree of






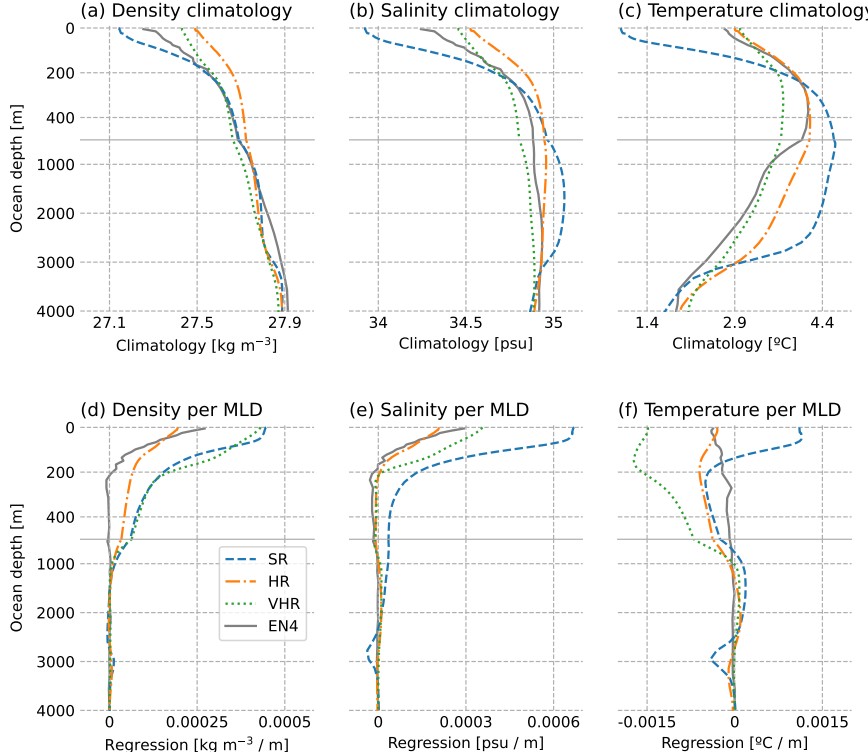

**Figure 3.** March vertical profiles for SR, HR, VHR, and EN4 1948-2023 in the Labrador Sea box (from Fig. 2). The first row shows sigma0 density (a), salinity (b), and temperature (c) climatologies. The second row shows sigma0 density (d), salinity (e), and temperature (f) regression coefficients with the March Labrador Sea MLD time series. The horizontal grey line divides the upper 500 m, where data has been zoomed in, from the lower 500 m.

agreement of the simulations with the observed profile, namely the root-mean-square-error (RMSE) and the correlation, both estimated in the vertical dimension. SR is the resolution with the best agreement with EN4 in terms of RMSE, while VHR has the best agreement in terms of correlation (Table 2). Overall, HR shows the poorest agreement with EN4, and it is also the less stratified resolution, which can explain why its MLD climatology is overly strong, as the threshold it has to overcome is smaller. This can also be seen in Fig. 3d, which shows the linear regression coefficients of the MLD time series with the vertical density

profile, as an additional diagnostic to compare the models with the observations. The density anomaly needed in the top ocean to generate one meter of MLD is much smaller in HR compared to the other two resolutions, although, interestingly, it stays closer to the observed regressed value. Somehow surprisingly, the largest differences with respect to the observed regressions happen for VHR, which had the most realistic climatological profile.

To gain further insights, the analysis is expanded to the temperature and salinity profiles. Even if the vertical density profile may suggest that SR is the model resolution in best agreement with the observational dataset near the surface, we can see

that it happens for the wrong reasons, as there is a strong error compensation between the biases in the salinity (Fig. 3b)





and temperature (Fig. 3c). When looking at those variables, VHR is consistently the resolution in best agreement with EN4 regarding both temperature and salinity profiles, also reflected in the correlation coefficients and RMSEs shown in Table 2. Interestingly, for all three resolutions, the density profile is clearly dominated by salinity, as inferred by the vertical profiles of the haline and thermal contributions to the Labrador Sea densities (Fig. A1), in which temperature has a minor influence and opposes the contribution from salinity.

Salinity regressions against MLD time series show that overall HR has the best agreement with EN4 (see Fig. 3e), with VHR being the second best, with overly salty regression values in the upper ocean. By contrast, SR overestimates the regression values at the surface by a factor of two, which might be related to the large negative bias in the corresponding climatological profile (Fig. 3b). The temperature regression against the MLD shows the largest qualitative discrepancies between models and observations (Fig. 3f). While observations show a moderate negative link between the mixing and local temperatures that is maximum at the surface and decreases monotonically with depth, both HR and VHR do show a negative link, but stronger in the subsurface ($\sim 200\,\mathrm{m}$). It is also worth noting that regression values for VHR are almost one order of magnitude higher than for HR and the observations. However, the largest discrepancies with respect to EN4 occur for SR, for which the regression values near the surface are of the opposite sign (i.e. positive instead of negative). This change in sign can be explained by the particularly strong local bias in sea ice, which is associated both with cold ocean conditions and reduced mixing. The inter-model differences revealed by the regression patterns for the mixed layer depth point to potential differences in the drivers of Labrador Sea mixing, which are investigated in the following.

## 3.2 Drivers of deep water mixing variability in the Labrador Sea

We first compare across resolutions the local atmospheric forcing exerted by the zonal wind, which in winter generally brings cold air masses from the continent. Figure 4a-c shows the correlation in time between December-March (DJFM) wind stress in the x-grid direction (zonal in mid-latitudes) over the North Atlantic and the time series of the Labrador Sea MLD in March. All the resolutions show a dipole-like pattern, with some qualitative and quantitative differences. In HR and VHR configurations, the pattern shows a strong positive correlation band going from approximately 40º N to 65º N and a negative correlation band south of 40º N. This dipolar wind structure has typically been associated with a positive North Atlantic Oscillation (NAO) phase (e.g. Ortega et al., 2012) and also matches the one from the first EOF of the zonal wind stress (not shown). This means that, for both resolutions, positive NAO phases tend to increase the mixing, while negative NAO phases tend to reduce it, as already shown in other studies (Patrizio et al., 2023; Ortega et al., 2021). To corroborate this, Fig. 4a-c also includes in contours the correlation of Labrador Sea MLD in March with the mean sea level pressure in DJFM, which shows a clear NAO-like dipole, with positive correlations over the Azores High and negative correlations over the Icelandic Low. Following the increase in the zonal wind, both HR and VHR show a cooling of the surface water masses through enhanced heat loss to the atmosphere (Fig. 4e-f), which makes them denser and thus promotes local mixing as shown by Kostov et al. (2019). In the case of SR, we observe a similar pattern for the wind stress to that in HR and VHR, but with weaker correlation coefficients that are also more confined to the basin's western side. The weaker correlations for the wind also result in weaker correlations for the surface heat fluxes, which are not significant in the west of the Labrador Sea, where the presence of sea ice precludes the local interactions



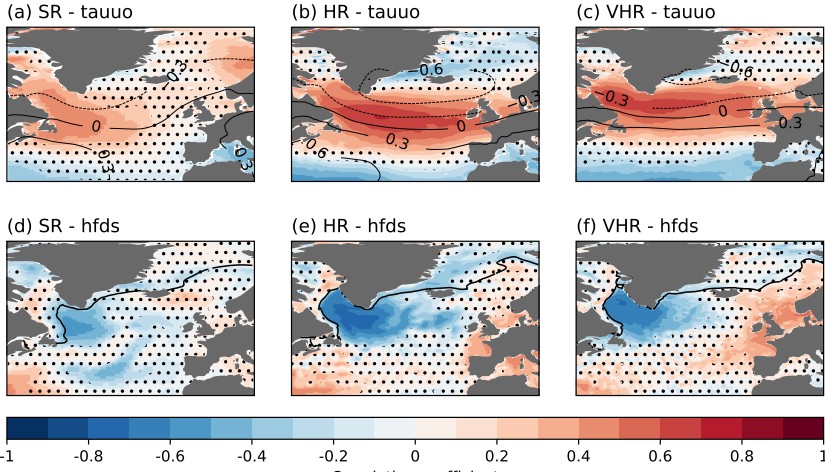

**Figure 4.** (a-c) Correlation of the Labrador Sea MLD in March with the DJFM eastward wind stress (colours) and the DJFM sea level pressure (contour lines) over the North Atlantic for SR, HR and VHR, respectively. Non-significant values are masked with dots. (d-f) The same but between the Labrador Sea MLD in March and the DJFM surface downward heat fluxes. In these three panels the thick black contour line shows the climatological line of 15% sea-ice concentrations in March.

with the atmosphere on the western side of the sea. This results in the impact of the NAO being less strong and widespread as for the other two resolutions. In all cases, the impact of atmospheric forcing through the zonal wind stress happens on a year-to-year scale. The next considered driver is associated with longer time-scale advective processes.



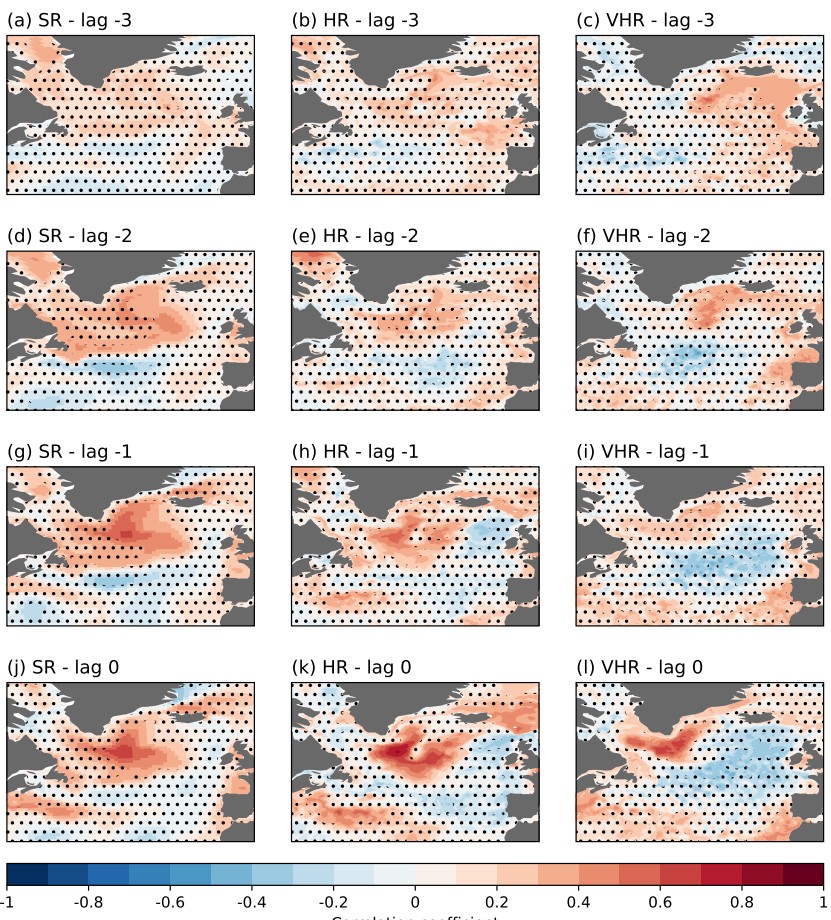

**Figure 5.** Correlation of March Labrador Sea MLD with yearly surface salinity, for SR (a, d, g, j), HR (b, e, h, k), VHR (c, f, i, l); lagged three years (a, b, c,), two years (d, e, f), one year (g, h, i), and no lag (j, k, l).



At all three resolutions, salinity has been shown to have an important role in explaining the surface density profiles as well
as the changes in the MLD in the Labrador Sea. We will now explore the origin of those salinity signals and how they are
linked to changes in the mixing by computing correlations between the annual surface salinity fields and the March MLD
index in the Labrador Sea (Fig. 5). When both variables are in phase, that is, at lag 0, all three resolutions show strong positive
correlations with salinity in the Irminger and the Eastern Labrador Seas, extending farther east in SR and HR, and westwards
into the Labrador Sea in VHR (Fig. 5j-l).

The subsequent lagged correlations allow us to track down the origin of the surface salinity signals. When lagging the
MLD by one year (Fig. 5g-i), SR and HR still show positive significant correlations in the same area, which suggests that
they build up over the years, probably due to positive feedback linked to the local salinity stratification conditions. VHR no
longer shows significant correlations in the Labrador Sea, but it does a bit further North near the Irminger Sea, where the other
two resolutions also show slightly higher correlation values than at lag 0, which suggests some salinity propagation. When
the MLD index lags by two years (Fig. 5d-f), the three resolutions show a positive significant correlation in the northeast of
the Irminger Sea, south of the Denmark Strait, further supporting the aforementioned propagation. Likewise, when the MLD
index lags by three years (Fig. 5a-c), we still see significant correlations in the same area for HR, and in the case of VHR, the
area of significant correlations is displaced to the East, extending from Iceland all the way to the British Islands. By contrast,
no significant correlations are found for SR in those regions. The propagation of salinity anomalies, more evident in VHR,
is probably explained by the mean transport of the subpolar gyre circulation. Similar correlation patterns have been produced
against the surface temperature fields (Fig. A2), but they do not show any clear advection of temperature signals into the
Labrador Sea for any resolution, which excludes temperature propagation as a key driver of Labrador Sea mixing.

Our results so far have shown differences across the three resolutions of EC-Earth3P in terms of vertical mixing, preconditioners and drivers of its variability, which could result in major differences regarding their link with the ocean circulation. In
the next section, we will therefore study this link with the AMOC.

## 3.3 Impact of resolution in the variability and meridional coherence of the Atlantic meridional overturning circulation

The mean state of the AMOC also varies with resolution. All analyses in this section are based on the volume streamfunction
in density space, which is more adequate to represent the contributions of deep water formation in the subpolar North Atlantic
than in depth space (Zhang, 2010; Foukal and Chafik, 2024). Furthermore, the Ekman transport has been removed from the
volume streamfunction to focus on its thermohaline component, which is more directly linked to the vertical mixing. When
computing the volume overturning stream function without the Ekman transport in sigma2-space, all the models show the
maximum value of the climatology around 55º N and 36.8 $\mathrm{kg\,m^{-3}}$ (Fig. 6, contour lines), but with different magnitudes. HR
and VHR show the strongest AMOC, with values greater than 20 Sv in both cases although slightly higher in HR. Meanwhile,
in SR the highest value is lower than 20 Sv. The three resolutions have the strongest variability of the AMOC situated at the
same latitude as the maximum but at a slightly higher density level (Fig. A3). Interestingly, despite having the weakest mean







**Figure 6.** Correlation of March Labrador Sea MLD with yearly averaged volume overturning stream function without the Ekman transport in sigma2-space, for SR (a, d, g, j), HR (b, e, h, k), VHR (c, f, i, l); stream function with no lag (a, b, c), lagged one year (d, e, f), two years (g, h, i), four years (j, k, l). Non-significant values are masked with dots. Contour lines show the climatologic volume overturning stream function without the Ekman transport in sigma2-space.



AMOC state of the three resolutions, SR shows higher variability than the rest, with HR and VHR exhibiting similar values. Therefore, the AMOC mean state and variability are largely comparable between HR and VHR, but differ in SR.

Newly formed dense waters in the Labrador Sea are usually propagated southwards along the DWBC (Ortega et al., 2021).
As these dense waters resulting from the vertical mixing move southward, they change the zonal density gradient, triggering a thermal wind balance response that leads to a general intensification of the AMOC. To explore whether this link is affected by resolution, Fig. 6 compares the response of the AMOC to changes in the Labrador Sea MLD in March across the three resolutions, using lagged correlations in which the Labrador Sea mixing always leads the AMOC. For the SR model, the lagged correlations of the overturning stream function in density space with the MLD show how, over lead times from zero to
four years, the significant correlations at the densest levels ($\geq 36.7\,\mathrm{kg\,m^{-3}}$, i.e., those formed by the deep mixing) progressively move southwards, reaching 20º N. However, the propagation pattern is different at finer resolutions. At lag 0, HR and VHR show much stronger correlations than SR in the subpolar latitudes (i.e., north of 45º N), where deep water mixing occurs. The main differences between HR and VHR emerge at subsequent lead times, when significant correlations progressively reach more southern latitudes, in line with a southward propagation of the AMOC signals. In the case of HR, a very slow
propagation is hinted, with the area of significant correlations reaching the 35º N latitude by the fourth lead year. Meanwhile, VHR shows a very quick drop of the correlation values already in the first lead year, with significant correlations limited to the 40-50º N latitudinal band. Interestingly, at lag 2, we observe a wide range of densities (i.e., 35 to 37 $\mathrm{kg\,m^{-3}}$) showing significant correlations south of 25º N, although the lack of coherence with the previous subpolar signals questions if they have propagated from the subpolar region, represent a local response or are simply a spurious correlation.

The southward propagation of the AMOC response to changes in the Labrador Sea deep mixing can be better captured if the density level is fixed, as this allows the temporal evolution to be visualised in a single plot. Figure 7a-c shows the correlation of Labrador Sea MLD in March with the annual AMOC at the 36.73 $\mathrm{kg\,m^{-3}}$ sigma2 level, as a function of lead-time with MLD leading. The largest correlations between AMOC and MLD happen at lag 0 and between 50º N and 65º N. Subsequent lead times show how the significant positive correlations move southward, although with notable differences across resolutions
regarding the timescales and the southward extent of the AMOC propagation. The SR shows a relatively slow propagation to about 45º N in about two to three years, with correlations becoming insignificant beyond that lead time and latitude. This slow propagation appears to be consistent with the advective propagation regime described by Zhang (2010), although they found it to reach 34º N. The propagation in HR shows two distinct timescales; there is a fast propagation within the first year to approximately 40º N, followed by a much slower one that reaches 35º N over eight years. The fast one could be linked to a
fast wave propagation, as even if the propagation pattern found in HR does not match the one described by Zhang (2010) it is consistent with recent results by Kostov et al. (2023). In the VHR model case, a much weaker cross-latitudinal link between the MLD and the AMOC is seen. Only a fast propagation as the one seen in HR occurs at this resolution, although with weaker correlations that also remain significant for a shorter period.

To further inspect the cross-latitudinal coherence of AMOC changes, which might not be necessarily driven by changes
in the Labrador Sea mixed layer, we have recomputed the same lagged-correlations but between the AMOC fixed at 55º N (where it exhibits the strongest climatological values) and the basin-wide AMOC streamfunction, both defined again at the





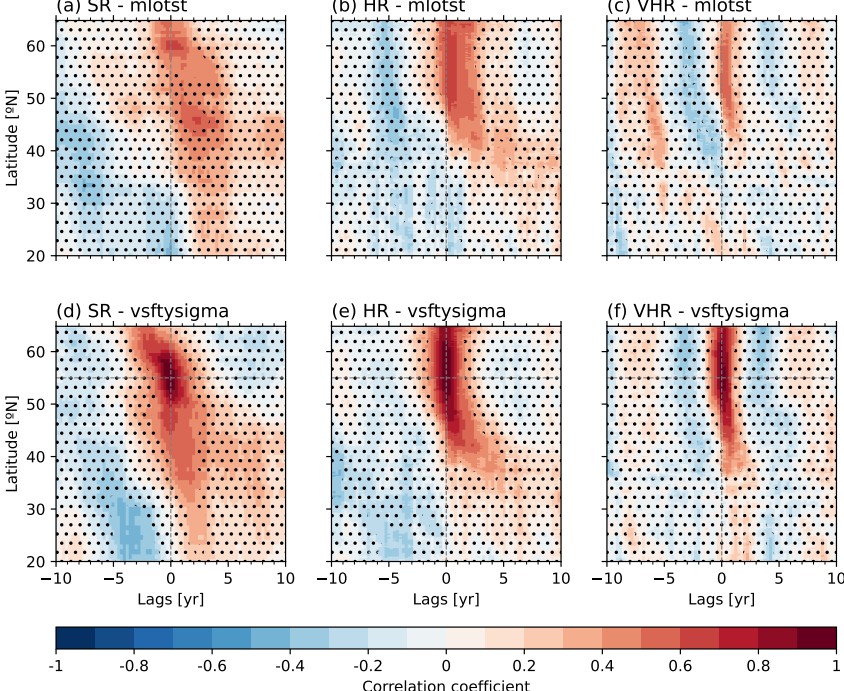

**Figure 7.** Correlation of monthly volume overturning stream function without the Ekman transport at $36.73 \, \mathrm{kg \, m^{-3}}$ sigma2 density level with March Labrador Sea MLD (a, b, c) and with itself at 55º N (d, e, f), for SR (a, d), HR (b, e), VHR (c, f). Non-significant values are masked with dots.

$36.73 \, \mathrm{kg \, m^{-3}}$ sigma2 level (Fig. 7d-f). As expected, this version has systematically higher correlation values than the previous one, which helps to better illustrate the gradual southward propagation of the AMOC changes. For SR we now see that the subpolar AMOC is connected with the AMOC at 20° N with a delay of 2-3 years. For HR, the main results are very similar to

those previously described for the correlations with the mixing, with a very clear fast propagation from the northern latitudes to 45º N, followed by a slowly paced propagation to 30º N over the next six years. In VHR we can now see that the strength of the correlations is comparable to that in HR and SR, in contrast with the correlations against the mixing. We also note for VHR an almost continuous band of significant correlations from the subpolar to the tropical latitudes, with a 1-2 year lag between 55° N and 37° N, and an almost instantaneous connection between 35º N and 20º N. This plot suggests that for VHR, other

influences than the Labrador Sea mixing are responsible for the southward propagation of AMOC changes, which is consistent with some recent studies suggesting that Labrador Sea water formation may not be a crucial driver of AMOC (Lozier et al., 2019; Zhang and Thomas, 2021).

The lagged correlations of the large-scale density field with the Labrador Sea MLD index give further information on how the signal produced by the mixing propagates along the western boundary, subsequently impacting the AMOC (Fig. 8). For

each grid point, we extract the maximum correlation in the depth range from 400 to 1400 m, that is where the higher change



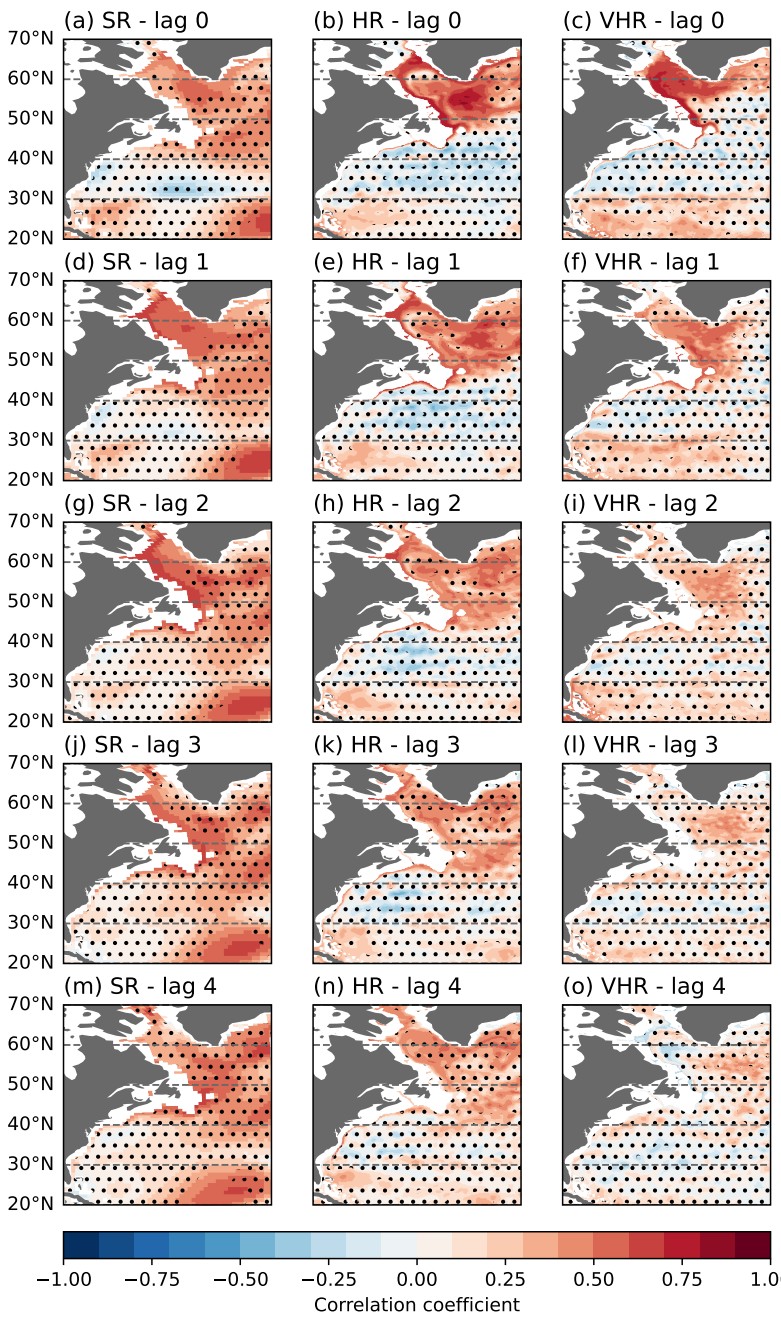

**Figure 8.** Maximum correlation between the Labrador Sea MLD index and the yearly sigma2 density fields from 400 m to 1400 m for SR (a, d, g, j,m), HR (b, e, h, k,n), VHR (c, f, i, l,o); stream function with no lag (a, b, c), lagged one year (d, e, f), two years (g, h, i), three years (j, k, l); four years (m,n,o). Non-significant values are masked with dots.





in density related to mixing happens (Fig. A4). To better examine the structure of the boundary current we also computed the lagged correlations between the Labrador Sea MLD index and the ocean densities along a zonal cross section located at 45° N, off the coast of Newfoundland (Fig. A5). Other cross-sections have also been considered, yielding similar results (not shown). At lag 0, all simulations show significant positive correlations in the interior Labrador Sea, but with differences across

resolutions in their location (more central in SR, flanked to the east in HR, and to the west in VHR). Inter-model differences are also found along the boundary currents. While in SR, only the Greenland Current shows clear and significant correlations, in HR, significant values (stronger than for the Labrador Sea Interior) extend from the Greenland Current to the Labrador Current and around the Grand Banks area down to 40° N, remaining significant until Cape Hatteras. In VHR, a distinct band of significant correlations extends from the Labrador Current all the way to Florida (∼ 30° N). These results thus describe a very

rapid (subyearly) propagation occurring both at HR and VHR, with the final latitude reached likely determined by the location where density anomalies in the interior Labrador Sea are formed. In VHR, significant correlations are also found further South, but they are stronger and span a much wider zonal extent, suggesting that they originate from a different process than the boundary propagation (e.g., a wind effect not accounted for when Ekman transport is removed).

At subsequent lags, further differences across resolutions are revealed. For SR, a slow propagation is hinted at, with sig-

nificant correlations gradually reaching 40° N at lag 1, 35° N at lag 2, and 25° N at lag 3. As expected from the coarser grid spacing and higher viscosity in SR, the DWBC is also wider and more zonally coherent than at higher resolutions. However, significant correlations in SR tend to remain more constrained to the coast and occur at shallower levels than in HR and VHR (Fig. A5). For HR, correlations remain significant along the boundary at all lags considered, reaching the southernmost tip of Florida at lag 3. This contrasts with the results of VHR, for which significant correlations are no longer found along the

boundary by lag 3. The greater persistence of significant correlations in HR, along with the gradual southward displacement of the latitudes with the strongest correlation values, is consistent with the slow propagation of AMOC signals found for this simulation in Fig. 7.

## 4 Discussion and conclusions

This paper explores whether and how the simulated mixed layer depth (MLD) in the subpolar North Atlantic, as well as its

main drivers and links with the Atlantic Meridional Overturning Circulation (AMOC), are affected by horizontal resolution. For that we use HighResMIP 1950-control experiments at three different resolutions of the coupled global model EC-Earth3P: an eddy-parametrised version (1º in the ocean mid-latitudes; SR), an eddy-present one (0.25º in the ocean mid-latitudes; HR), and an eddy-rich one (1/12º in the ocean mid-latitudes; VHR); all of them developed during PRIMAVERA project and following the HighResMIP protocol (Moreno-Chamarro et al., 2024).

Important differences between the different resolutions have been found. The main findings of our study are summarised as follows:

- The Labrador Sea is the Northern Hemisphere region with the deepest mixed layer in all model versions, with VHR's climatological values showing the best agreement with EN4 observations both in terms of magnitude and spatial extent.



HR largely overestimates the Labrador Sea MLD, and SR underestimates it, partly due to a positive sea ice bias. None
of the configurations, however, show ocean deep mixing in the Nordic Seas, as found in the observations, which can be
linked to an excess of regional sea ice.

–   The more realistic climatological value of the Labrador Sea MLD in VHR is consistent with an improved representation
of the local stratification with respect to the other two resolutions, and more in particular to a better representation of the
vertical temperature and salinity profiles.

–   The atmospheric forcing on the Labrador Sea mixed layer depth is very similar in VHR and HR configurations and is
ultimately linked to the North Atlantic Oscillation (NAO). A positive NAO phase is found to drive an increase in MLD
mixing and the response occurs for negative NAO phases. In SR, the atmospheric forcing is weaker and more constrained
to the basin's western side, which could be partly due to a shielding effect of the overly extended sea ice on the ocean.

–   Surface salinity is also found to be an important contributor to the variability of the Labrador Sea mixing at all resolutions,
facilitated by a positive feedback that enhances the persistence of the local salinity signals. Differences inter-model
differences emerge regarding the origin of the surface salinity signals. In VHR, Labrador Sea mixing is preceded by a
slow propagation of surface salinity signals from the eastern flank of the subpolar gyre into the Labrador Sea. Such a
distant propagation is absent in both HR and SR.

–   Labrador Sea MLD imprints on the AMOC show three distinct behaviours both in terms of latitudinal extension and per-
sistence of the AMOC changes. While VHR shows an almost instantaneous AMOC to the mixing, of limited latitudinal
reach and persistence, SR shows a more gradual and persisting response of similar latitudinal reach, and HR shows a first
rapid response down to 40°N, followed by a slow response that gradually reaches the Tropics. An additional analysis
of the AMOC's cross-latudinal coherence shows a significant connection in VHR between the subpolar AMOC and the
subtropics, not evidenced in the AMOC response to Labrador Sea MLD, which suggests the presence of other important
drivers of AMOC variability.

–   An analysis of the link between the Labrador Sea MLD and the densities across the Deep Water Boundary Current
confirms a very fast propagation of Labrador Sea density signals along the boundary current in VHR, a combination
of fast and slow propagation in HR, and a slow propagation in SR. These boundary densities are expected to drive,
through thermal wind balance, a latitudinally coherent AMOC response that reaches the subtropics. In VHR, however,
the boundary signals are rather weak when they reach Cape Hatteras, which might explain why they are unable to
generate an AMOC response in the Subtropics.

This study investigated the differences in the mixed layer depth, its drivers, and its impact on the AMOC across resolutions.
It therefore makes sense to discuss how well every resolution compares to observations; as a way to elucidate if VHR is actually
more realistic, thus justifying its substantially higher computing costs to carry on similar studies. However, the comparison to
observations was hindered by different factors. The most important factor is that our simulations correspond to a period of fixed



1950 radiative forcing conditions, while observations describe an evolution that was shaped by transient radiative forcings. Also the availability and quality of oceanic observations near the year 1950 are very limited, particularly for the subsurface. Indeed, the number of observations has dramatically increased in the last 30 years (Gould et al., 2013), thanks to the deployment of satellites, the launch of in-situ measurement projects like Argo (started 20 years ago; Roemmich et al., 2009), and transport-
measuring arrays like RAPID (started 20 years ago; Moat, 2023) at 26º N and OSNAP (started 10 years ago; Fu et al., 2023) in the subpolar region. However, this period is substantially warmer than 1950, and thus not representative of its mean climate. For this reason, we limited our comparison with observations to variables and regions with reasonably good data coverage around the 1950s. Another major limitation related to the study of control experiments concerns the evaluation with observations of temporal features and covariances between variables, which include externally forced signals in the observations that may not
be possible to remove with a polynomial detrender. Moreover, the limited time span of mass-transport observations from the OSNAP and RAPID arrays, along with the sparse spatial information they provide, substantially hinders our understanding of the real-world linkages between the mixed layer depth and the AMOC (Jackson et al., 2022; Frajka-Williams et al., 2023). This same problem hampers our knowledge of how coherently the AMOC changes across latitudes. Extending sustained AMOC measurements to other latitudes, and improving the realism of ocean reanalyses for dynamical large-scale variables is therefore
essential to fill that knowledge gap.

Multimodel analyses are also important for building confidence in the impact of model resolution and its potential added value, in particular for processes where observations are sparse. Results that are consistent across models are more likely to be reliable, whereas model-dependent results indicate higher uncertainty. In this sense, previous studies suggest that while our results might be model-dependent, others are consistent. Koenigk et al. (2021) explores the impact of resolution on the
mixed layer depth, and includes HadGEM3-GC31-HH, which shares the same ocean component (NEMO3.6) of EC-Earth3P-VHR configured at the same ocean resolution. However, the MLD from HadGEM3-GC31-HH is more comparable to our HR configuration than to VHR, also showing stronger convection in the Nordic Sea, which is underestimated in our experiments. AMOC latitudinal coherence (defined as the connection between the AMOC anomalies across latitudes) has been shown in Fig. S1 of Roberts et al. (2020) to vary across models and resolutions. The propagation pattern we find in VHR is similar to
the one they show for HadGEM3-GC31-HH, which suggests that eddy-rich models have a weak AMOC coherency between subpolar and subtropical latitudes.

It is also important to consider that the 76 years considered in our analyses may be insufficient to sample the decadal and multidecadal variability, and constrain accurately the statistical significance of the covariances explored. However, this is a problem that derives from the HighResMIP protocol itself, which considers 100-year control simulations as a compromise
to assess internal variability at interannual scales, while minimising the computational and data storage needs. Longer model integrations or multi-ensemble simulations are therefore desirable to improve our understanding of decadal variability, which might be more affordable by coordinating efforts between different modelling centres, such as is currently planned within the European project EERIE and the next phase of HighResMIP. More generally, these joint multi-model efforts are also important to shed new light on the key role played by mesoscale ocean eddies on the climate, including its future response to the projected
greenhouse gas emissions.





*Code and data availability.* EN.4.2.2 data is available from https://www.metoffice.gov.uk/hadobs/en4/index.html. HadISST2 data is available from https://www.metoffice.gov.uk/hadobs/hadisst/. SR (2019, EC-Earth3P) and HR (2018, EC-Earth3P-HR) data is available from ESGF (https://esgf-index1.ceda.ac.uk/search/cmip6-ceda/). VHR (EC-Earth3P-VHR) data will be published in ESGF, meanwhile, the data could be shared through an FTP upon request.

Used ESMValTool recipes and diagnostics will be made available in the revised version.



## Appendix A: Supplementary figures

$$\text{SigmaS} = b\left(S - \overline{S}\right) \tag{A1}$$

$$\text{SigmaT} = -a\left(T - \overline{T}\right) \tag{A2}$$

$$\text{Sigma} = \left(sigma0 - \overline{sigma0}\right) = \text{SigmaS} + \text{SigmaT} \tag{A3}$$

where $a$ is the thermal expansion coefficient, and $b$ is the haline contraction coefficient.

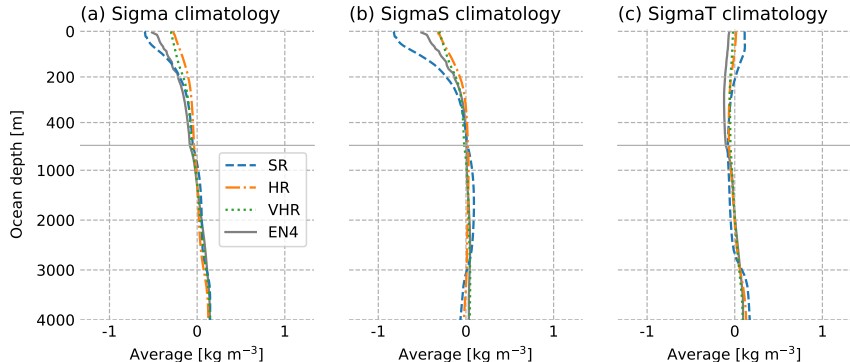

**Figure A1.** March vertical profiles for SR, HR, VHR, EN4 1948-2023 in the Labrador Sea box for relative density sigma0 (a), salinity contribution to relative density sigma0 (b), and temperature contribution to relative density sigma0 (c) climatologies. These variables have been computed following Eqs. (A1-A3).



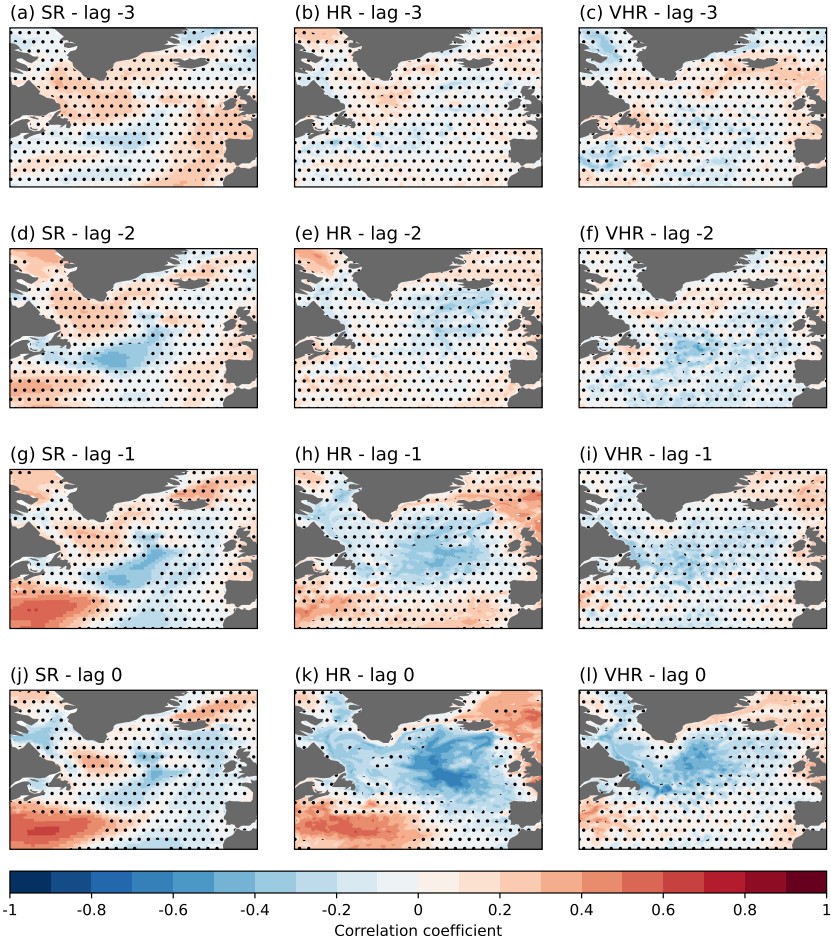

**Figure A2.** Correlation of March MLD time series with yearly surface temperature, for SR (a, d, g, j), HR (b, e, h, k), VHR (c, f, i, l); lagged three years (a, b, c), two years (d, e, f), one year (g, h, i), and no lag (j, k, l).





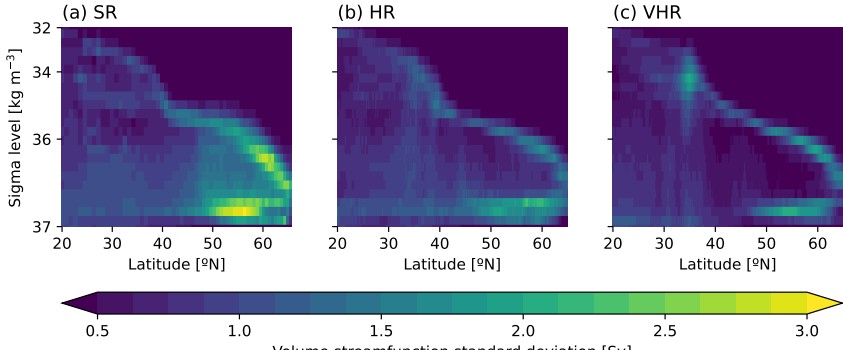

**Figure A3.** Standard deviation of the volume overturning stream function without the Ekman transport in sigma2-space, for SR (a), HR (b), and VHR (c).



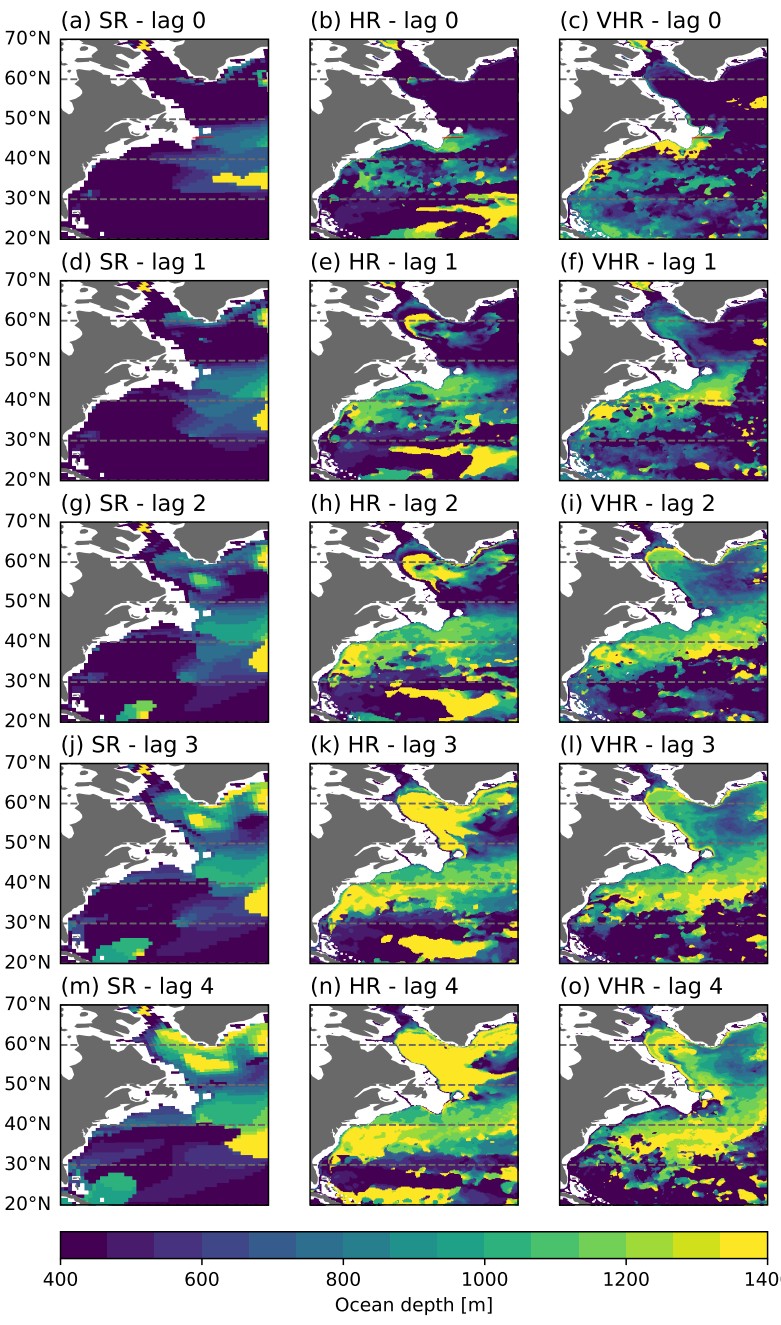

**Figure A4.** Depth of maximum correlation between 400 m and 1400 m of yearly density with MLD. Complementary for Fig. 8. The red line (a, b, c) shows the position of the depth vs longitude sections in Fig. A5 were computed.



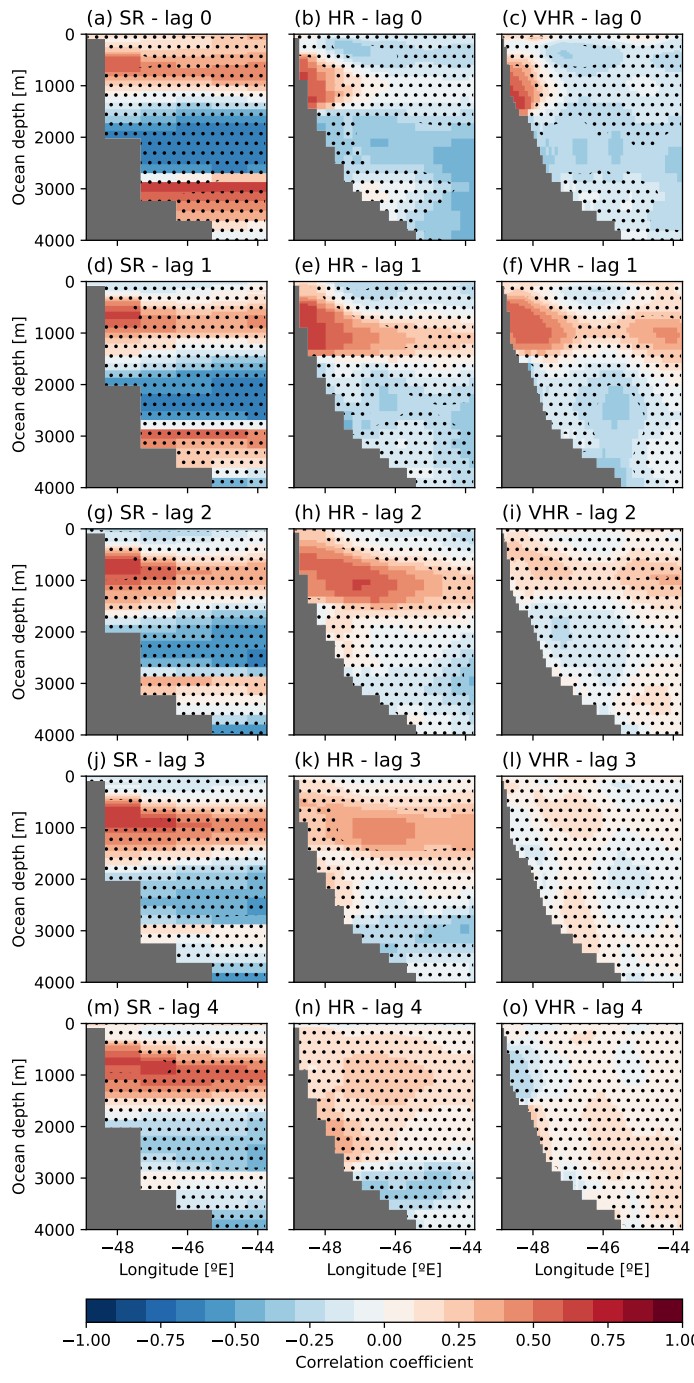

**Figure A5.** Correlation of March MLD time series with yearly sigma2 at 45.3° N, for SR (a, d, g, j), HR (b, e, h, k), VHR (c, f, i, l); no lag (a, b, c), one-year lag (d, e, f), two-year lag (g, h, i), three-year lag (j, k, l), and four-year lag (m, n, o). The section is shown with a red line in Fig. A4(a-c).



*Author contributions.* EMM carried out the analysis and wrote the manuscript. AF, EMC, and PO suggested analysis and gave inputs to the manuscript. PAB and DK post-processed and cmorized the model data. SLT gave support using ESMValTool and made improvements for memory usage and performance in the tool needed for the analysis. MSC downloaded and prepared for processing observations, SR, and HR data. EMC ran the simulations.

*Competing interests.* The authors declare that they have no conflict of interest.

*Acknowledgements.* This publication is part of the EERIE project (Grant Agreement No 101081383) funded by the European Union. Views and opinions expressed are however those of the author(s) only and do not necessarily reflect those of the European Union or the European Climate Infrastructure and Environment Executive Agency (CINEA). Neither the European Union nor the granting authority can be held responsible for them.

EMM received funds from grant PRE2021-097163 funded by MCIN/AEI/10.13039/501100011033 and by ESF Investing in your future. EMM, EMC, and AF received funds from grant PID2020-114746GB-I00 funded by MICIU/AMEI/10.13039/501100011033.



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
