# Peer review of "Effect of horizontal resolution in North Atlantic mixing and ocean circulation in the EC-Earth3P HighResMIP simulations"

_EGUsphere, 2024_

## Referee Comment (RC2)

The authors performed simulations with the use of the climate model EC-Earth3P with 3 different spatial resolution, which are eddy-parametrised (SR), eddy-Permitting (HR) and eddy-rich (VHR). The goal is to examine the resolution impact on the oceanic mixing processes, their drivers and AMOC. They firstly compared the simulated mixed layer depth distribution in the North Atlantic, vertical profiles in density/temperature/salinity in Labrador Sea with observations, showing the best performance of the VHR. They then show the resolution effect on the links of North Atlantic westerlies and surface salinity with the Labrador Sea mixed layer depth, as well as the link between Labrador Sea mixed layer depth and AMOC. The authors did a good job. Their work highlight the importance of using high resolution models to accurately capture realistic ocean properties and processes associated with AMOC. The manuscript is well-written, and the conclusions are generally supported by the presented analysis. I would recommend minor revisions for this stage.

Specific comments:

1 The authors focus on the Labrador Sea mixing and its connect with AMOC, as this region has been considered a key region affecting AMOC. Though it is true that all experiments show a deeper MLD in the Labrador Sea than other deep water formation regions, it doesn't mean the Labrador Sea processes are more important than the Irminger and the Nordic Sea in modulating the AMOC. For example, Ma et al (2024) shows that the Irminger basin could be the most effective region leading to AMOC changes though MLD is the deepest in the Labrador Sea. I would suggest to perform a lagged correlation between AMOC indices and mean surface density in all the three deep water formation sites, to first check which area is the key. An example is Fig. 5 in Shi and Lohmann (2016)

Ma, Qiyun, et al. "Revisiting climate impacts of an AMOC slowdown: dependence on freshwater locations in the North Atlantic." Science Advances 10.47 (2024)

Shi, X., & Lohmann, G. (2016). Simulated response of the mid-Holocene Atlantic meridional overturning circulation in ECHAM6-FESOM/MPIOM. Journal of Geophysical Research: Oceans, 121(8), 6444-6469.

Minor comments

2. How the simulated AMOC compared to modern estimation? Is the VHR also better in simulating AMOC strength/streamfunction than other two resolutions?

3 In Section 3.1 The authors show that the VHR behaves the best in simulating the vertical profile of Labrador Sea properties, is this improvement mostly related to more accurate eddy effects or more realistic presentation of ocean properties and processes,

e.g., topography, currents...

4. Regarding the different propagation speed in surface water in VHR versus other setups, and the differences in how the mixing propagates and impacts the AMOC, what kind of role is played by the meso scale eddies and high-resolution topography here?

5. Caption of Figure 7, please indicate who leads whom when lag >0.

6. Caption of Figure 8, if I understand it correctly, the figure is for correlation between MLD and density. Is the "stream function" a typo here?

---

## Author Comment (AC1)

**General comments**

In this paper, the authors investigate the effect of horizontal resolution on the North Atlantic mixed layer depth and its link to the AMOC. Their analysis is based on three simulations performed with the EC-Earth3 climate model.

Overall, the paper is well-structured and well-written. The analysis is sound, and results are mainly descriptive. The paper could benefit from a more thorough discussion of the broader implications of the findings.

We have added the following paragraph just after listing the conclusions:

"These results show different behaviour for the ocean circulation and its driving processes in the North Atlantic across resolutions. Therefore, further research is needed to confirm if similar differences are identified in other climate models, to thus determine if eddy-resolving models consistently bring new regimes of variability that could challenge our current understanding of the future changes to be experienced by the AMOC, which predominantly come from models with eddy-parameterized oceans."

My comments are mainly orientated to improve the presentation of the results, particularly the figures along with a few minor suggestions to improve the text.

We thank the Referee for dedicating time to review our paper and for the constructive feedback. The suggested changes help for a better understanding of the work improving its scientific quality. Answers to each comment can be found below.

**Specific comments on figures**

The manuscript includes several geographical references. While some (e.g., Labrador Sea, Irminger Sea, Nordic Seas) are familiar, others (e.g., Cape Farewell, Cape Hatteras) may not be as well-known to all readers. I recommend adding these locations to one of the maps.

Thank you for the suggestion. We have added a new figure to the appendix with all these locations. We have also added a reference to the figure at the end of the Methods section.

[Figure]

*Supporting Figure 1 (added as Fig A1): Map including geographical references that are relevant for the article.*

F1, F2, FA3. I suggest changing the colorbar of these figures because it is difficult to distinguish the differences, particularly for values below 1000m. Besides, it is hard to see the black line of the climatological sea ice superimposed to the dark contours.

We have tried to improve the visibility of the figures reducing the binning and using magma color gradient instead of viridis. We did not find many additional options of perceptually uniform sequential colormaps, which we prefer as they are colorblind friendly. We note that F1 and F2 have been expanded to occupy the full text width for better visualization (mainly affects the font size from the preprint version). See examples below for the resulting F1 and F2:

[Figure]

*Supporting Figure 2 (replaces Fig 1): March MLD climatology for SR (a), HR (b), VHR (c), and EN4 1940-1960 (d). The black contour line shows the climatological March sea-ice concentration at 15 \%, for the same dataset in (a-c), and HadISST2 1940-1960 in (d).*

[Figure]

*Supporting Figure 3 (replaces Fig 2): March standard deviation of MLD for SR (a), HR (b), VHR (c), and EN4 1948-2023 (d). The box used to compute MLD time-series and vertical profiles is shown in (a).*

Table 2. I am a bit confused with the information that T2 provides. What exactly does the correlation represent? If it shows the correlation of vertical profiles (model vs. observations), the high values may not be surprising, as they primarily reflect the expected stratification (densest waters at the bottom, lightest at the surface). For instance, the 0.987 correlation for VHR is only a slight improvement over 0.976 for SR. Moreover, visually, Fig. 3a suggests that SR may outperform HR and VHR in some aspects.

The reviewer is correct, T2 shows the correlation of time averaged vertical profiles across the vertical dimension of model vs observations. We agree that high correlation values are expected, because stratification has a smooth nature. We also acknowledge that correlation differences between configurations are small, so they need to be interpreted with caution, as not all features will be better captured by the same model. However, we note that visual inspection can be deceptive when drawing conclusions. That is why we use the metrics to evaluate which model configurations are better at representing stratification. The important aspect is that we used two different metrics (correlation and RMSE) and looked at three different key variables to assess if the same model configuration stands out in all cases. Interestingly, VHR consistently shows the best performance, showing the best metric for all cases but the RMSE value of the density profile, in which it's ranked second after SR. This gives us confidence to argue that VHR is still superior to SR, for which the temperature and salinity profiles are substantially worse, which indicates its density profile is well represented, but for the wrong reasons.. We now acknowledge in the text that for the particular case of correlation of density profiles, SR has similar peformane to VHR. For that we have updated the sentence in the manuscript to say: "VHR has the best agreement in terms of correlation, followed closely by SR"

F4, F5, F6, F7, F8, FA2, FA5. I suggest plotting dots when the values are significant instead of the other way around.

We understand that plotting the dots in the significant areas is the most common practice and what many readers will expect, which can lead to confusion.. However, we believe that masking the non-significant areas with the dots strongly improves the visibility of the significant values, which, in the end, is what we will analyse. In particular, adding dots over significant areas make them appear darker than they really are (i.e. with higher correlation), as we have illustrated in Supporting Figure 4 (equivalent to Fig. 7) below. For that reason, we prefer to keep the representation of significant areas as it was. We have now modified the Figure captions to say "Non-significant values are masked with dots to improve the visibility of the significant regions".

[Figure]

*Supporting Figure 4 (equivalent to Fig 7): Correlation of monthly volume overturning stream function without the Ekman transport at 36.73 kg m⁻³ sigma2 density level with March Labrador Sea MLD (a, b, c) and with itself at 55° N (d, e, f), for SR (a, d), HR (b, e), VHR (c, f). When lag > 0 March Labrador Sea MLD (a, b, c) or AMOC at 55° N (d, e, f) leads. Significant values are masked with dots.*

FA4. The red line in these plots is not distinguishable.

We have made the line thicker so it can be seen better. We have also updated the colorbar to magma and line's color to black to be consistent with the other plots.

**Minor comments and typos**

In several places, the cited papers are not properly ordered eg. L48, L64, L65, L76, L194, L238

Solved.

L13. 'highest' instead of 'higher'

Changed.

L96-97. Should be: Haarsma et al. (2020) and Moreno Chamarro et al. (2024).

Changed.

L203. 'less stratified resolution' sounds strange to me. I suggest 'the model version which represents the weakest vertical stratification'.There are other places in the manuscript referring to the model version as resolution alone that can be improved.

> Thank you. We have changed that to "the model configuration which represents the weakest vertical stratification". We have made similar improvements in the manuscript when referring to SR, HR or VHR as resolution.

L330. 'highest' instead of 'higher'.

> Changed.

L364. 'partly due to a positive sea ice bias'. This is not actually demonstrated in the current study, but it is so far a hypothesis.

> We have modified the statement to say: "an underestimation that is connected to a negative sea ice bias in that model configuration".

L371-372. 'A positive NAO phase is found to drive an increase in MLD mixing and the response occurs for negative NAO phases.' I don't understand this sentence, please rephrase it.

> We have rephrased it the following way: "Positive NAO phases are found to enhance the Labrador Sea mixing, while negative NAO phases reduce the mixing there."

L375-376. 'Differences inter-model differences'. Please, delete the first 'differences'.

> Deleted

L381. What does 'an almost instantaneous AMOC to mixing, of limited latitudinal reach and persistence' mean? I don't understand this sentence.

> We have split it in two with the next sentence and rephrased it in the following way: "VHR shows an almost instantaneous response of the AMOC to the MLD changes in the Labrador Sea, a response that has limited latitudinal reach and persistence in time. Meanwhile, SR".

---

## Author Comment (AC2)

The authors performed simulations with the use of the climate model EC-Earth3P with 3 different spatial resolution, which are eddy-parametrised (SR), eddy-Permitting (HR) and eddy-rich (VHR). The goal is to examine the resolution impact on the oceanic mixing processes, their drivers and AMOC. They firstly compared the simulated mixed layer depth distribution in the North Atlantic, vertical profiles in density/temperature/salinity in Labrador Sea with observations, showing the best performance of the VHR. They then show the resolution effect on the links of North Atlantic westerlies and surface salinity with the Labrador Sea mixed layer depth, as well as the link between Labrador Sea mixed layer depth and AMOC. The authors did a good job. Their work highlight the importance of using high resolution models to accurately capture realistic ocean properties and processes associated with AMOC. The manuscript is well-written, and the conclusions are generally supported by the presented analysis. I would recommend minor revisions for this stage.

We thank the Referee for the time dedicated to read the manuscript and for the constructive comments. The suggested changes have improved the clarity of the article. Answers to each comment can be found below.

**Specific comments:**
The authors focus on the Labrador Sea mixing and its connect with AMOC, as this region has been considered a key region affecting AMOC. Though it is true that all experiments show a deeper MLD in the Labrador Sea than other deep water formation regions, it doesn't mean the Labrador Sea processes are more important than the Irminger and the Nordic Sea in modulating the AMOC. For example, Ma et al (2024) shows that the Irminger basin could be the most effective region leading to AMOC changes though MLD is the deepest in the Labrador Sea. I would suggest to perform a lagged correlation between AMOC indices and mean surface density in all the three deep water formation sites, to first check which area is the key. An example is Fig. 5 in Shi and Lohmann (2016)

Ma, Qiyun, et al. "Revisiting climate impacts of an AMOC slowdown: dependence on freshwater locations in the North Atlantic." Science Advances 10.47 (2024)

Shi, X., & Lohmann, G. (2016). Simulated response of the mid-Holocene Atlantic meridional overturning circulation in ECHAM6-FESOM/MPIOM. Journal of Geophysical Research: Oceans, 121(8), 6444-6469.

Thank you. This is a good point. We have replicated the analysis in Figure 7 for the three deep water formation regions of the Northern Hemisphere, using theboxes in the figure below, added as Fig A1 (including also information asked by RC1). We took a big box covering all the Nordic seas from 15º W 65º N to 20º E 80º N; the same box we use for the MLD in the manuscript but cut at 60º N to isolate the Labrador Sea; and a box starting at Cape Farewell until 67º N and 25º W for the Irminger Sea.

[Figure]

*Supporting Figure 1 : Map including geographical references and boxes used for Labrador Sea (orange), Irminger Sea (blue), and Nordic Seas (purple).*

We compute lead-lagged correlations in the given boxes between MLD and AMOC (to check the driving role of mixing in each of the boxes; Sup. Fig. 2) and the same between surface sigma0 and AMOC (as suggested by the reviewer). However, we note that surface density anomalies do not necessary lead to deep mixing anomalies in that same region, as they can be transported to another region where they are mixed vertically; Sup. Fig. 3). Any of the regions or variables show a correlation of similar shape and strength to the AMOC correlation with itself (Fig 7d-f). This may indicate that the changes on the AMOC can be driven by different sources (mixing in several regions, wind forcing…) as we may expect.

When comparing the correlations of MLD in Labrador and Irminger seas, we find similar patterns for HR and VHR, which also are similar to those in Fig 7b-c. In case of SR, the area where the correlation between tha variables is statistically significantant is much bigger in the Labrador Sea box, which is also similar to that one in Fig 7a. This justifies the box we took in the manuscript (Fig. 2), which extends further north of the Labrador Sea box defined before. Regarding Nordic seas, there is only SR shows a connected significant propagation pattern that reach low latitudes. However, the pattern starts at negative lags of 3 years, which implies an AMOC leading the mixing there. This may discard this region's mixing change from being driver of the changes in the AMOC.

When checking the correlation with sigma0, we only find a more clear pattern of cause-effect in the Irminger Sea for SR. For HR the correlation in Labradror Sea is similar to the one in Fig 7b. For VHR, the correlation in Irminger Sea is similar to that ibe in Fig 7c. Note that we discuss Labrador sea as a mixing area, where the MLD grows, but we also include the salinity anomalies brought from the eastern SPG as a driver of mixing (Fig 5), which can be related to the correlation of sigma0 - AMOC in Irminger Sea. For the Nordic seas we have similar results to those from the correlation of MLD with the AMOC.

[Figure]

*Supporting Figure 2: Correlation of monthly volume overturning stream function without the Ekman transport at 36.73 kg m$^{-3}$ sigma2 density level with March MLD at Labrador Sea (a, b, c), Irminger Sea (d, e, f), and Nordic Seas (g, h, i). For positive lag values, March MLD leads. Non-significant values are masked with dots to improve the visibility of the significant regions*

[Figure]

*Supporting Figure 3: Correlation of monthly volume overturning stream function without the Ekman transport at 36.73 kg m⁻³ sigma2 density level with March surface sigma0 at Labrador Sea (a, b, c), Irminger Sea (d, e, f), and Nordic Seas (g, h, i). For positive lag values, March MLD leads. Non-significant values are masked with dots to improve the visibility of the significant regions*

**Minor comments**

How the simulated AMOC compared to modern estimation? Is the VHR also better in simulating AMOC strength/streamfunction than other two resolutions?

We think the comparison is not so simple, as the estimations from observations are quite recent, and we are using 1950-control. Moreover, observations are only available from few arrays, which do not give a proper streamfunction structure across all latitudes. Checking Frajka-Williams et al 2019, we see that the three model configurations simulate weaker AMOC than the estimated one, both at 16º N and 26.5º N. However, the models simulate the maximum AMOC at different latitudes as described in other works (e.g. Danabasoglu et al., 2016). On first comparison VHR is not performing better the mean state, when comparing values at 16º N and 26.5º N, but as we have no estimations to where its maximum is (about 32º N in VHR, see Sup. Fig. 4) we cannot say that the AMOC at all latitudes.

MOVE 16º N

Similar values for the 3 model configurations, being all of them lower than the average of the observations.

Frajka-Williams et al., 2019

> Over the period February 2000–June 2018, the mean and standard deviation of the daily values are 18.0±5.8 Sv

Experiments (averge ± Standard deviation of the annual values)

> SR: 14.1 ± 0.9 Sv

> HR: 14.9 ± 0.9 Sv

> VHR: 14.8 ± 0.9 Sv

RAPID 26.5º N

Frajka-Williams et al., 2019

> Over the April 2004–February 2017 observational record, the mean and standard deviation of the overturning transport is 17.0±4.4 Sv

Experiments (averge ± Standard deviation of the annual values)

> SR: 15.8 ± 1.1 Sv

> HR: 15.4 ± 0.9 Sv

> VHR: 15.0 ± 0.9 Sv

32º N

Experiments (averge ± Standard deviation of the annual values)

SR: 14.5 ± 1.1 Sv

HR: 15.6 ± 1.0 Sv

VHR: 17.4 ± 1.2 Sv

Comparing with reanalysis is also tricky as the resolved AMOC is constrained to the resolution of the used model and its biases, e.g. Jackson et al. 2019. Therefore, we think that this comparison does not give us clear information about which model configuration/resolution is reproducing better the AMOC.

[Figure]

*Supporting Figure 4: Climatological volume overturning stream function in z-space, for SR (a), HR (b), and VHR (c).*

Eleanor Frajka-Williams, Atlantic Meridional Overturning Circulation: Observed Transport and Variability, Frontiers in Marine Science, Volume 6, 2019,
https://doi.org/10.3389/fmars.2019.00260

Gokhan Danabasoglu, North Atlantic simulations in Coordinated Ocean-ice Reference Experiments phase II (CORE-II). Part II: Inter-annual to decadal variability, Ocean Modelling, Volume 97, 2016, https://doi.org/10.1016/j.ocemod.2015.11.007

Jackson, L. C., Dubois, C., Forget, G., Haines, K., Harrison, M., Iovino, D., et al. (2019). The mean state and variability of the North Atlantic circulation: A perspective from ocean reanalyses. Journal of Geophysical Research: Oceans, 124, 9141–9170. https://doi.org/10.1029/2019JC015210

In Section 3.1 The authors show that the VHR behaves the best in simulating the vertical profile of Labrador Sea properties, is this improvement mostly related to more accurate eddy effects or more realistic presentation of ocean properties and processes, e.g., topography, currents…

It is difficult to attribute the improvements to any of the specific changes in the model configuration or related processes, as they come all together. It must be remarked that VHR configuration is not high enough to resolve all eddies in the Labrador Sea, where the model resolution would need to be finer than 1/12º (see the map in Hallberg 2013). However, VHR resolution allows resolving mesoscale eddies south of the Labrador Sea, where they are key for the transport of heat and salinity.

We have added the following to the manuscript:
"The better representation of the vertical profile in VHR may be due to different factors. Although VHR does not have the resolution capable of resolving eddies explicitly in the Labrador Sea (a resolution of 1/16º would be necessary; Hallberg 2013), it has a fine enough resolution to do it south of the Labrador Sea, where mesoscale eddies are key for the transport of heat and salinity to the Labrador Sea. In addition, better topography and resolution of boundary currents and the improved air-sea interaction, between many other processes, may also influence the correction of these biases."

Robert Hallberg, Using a resolution function to regulate parameterizations of oceanic mesoscale eddy effects, Ocean Modelling, Volume 72, 2013, https://doi.org/10.1016/j.ocemod.2013.08.007

Regarding the different propagation speed in surface water in VHR versus other setups, and the differences in how the mixing propagates and impacts the AMOC, what kind of role is played by the meso scale eddies and high-resolution topography here?

As before, it is not easy to attribute the origin in the observed changes. The improvements in the resolution and topography allow having narrower (or more confined) deep western boundary currents and also solving the boundary waves, such as the Kelvin waves, which may explain the faster propagation of the signal (Getzlaff et al. 2005).

Getzlaff, J., C. W. Böning, C. Eden, and A. Biastoch (2005), Signal propagation related to the North Atlantic overturning, Geophys. Res. Lett., 32, L09602, doi:10.1029/2004GL021002.

Caption of Figure 7, please indicate who leads whom when lag >0.

Done, added the sentence "For positive lag values, March Labrador Sea MLD (a, b, c) or AMOC at 55º N (d, e, f) leads."

Caption of Figure 8, if I understand it correctly, the figure is for correlation between MLD and density. Is the "stream function" a typo here?

Corrected, it was a typo.